Transcriptome analysis of Sonneratia caseolaris seedlings under chilling stress

Yang Yong 1
Zheng Chunfang 2
Zhong Cairong 3
Lu Tianxi 1
Gul Juma 1
Jin Xiang 1
Zhang Ying Zhangyingred@yahoo.com 1
Liu Qiang hnsylq@163.com 1
1 Ministry of Education Key Laboratory for Ecology of Tropical Islands, Key Laboratory of Tropical Animal and Plant Ecology of Hainan Province, College of Life Sciences, Hainan Normal University , Haikou , China
2 National and Local Joint Engineering Research Center of Ecological Treatment Technology for Urban Water Pollution, College of Life and Environmental Science, Wenzhou University , Wenzhou , Zhejiang , China
3 Hainan Academy of Forestry, Hainan Mangrove Research Institute , Haikou , Hainan , China
Kaminaka Hironori
Electronic publication date: 2021 Jun 3
Publication date: 2021
Volume: 9
Electronic Location ID: e11506
Received 2021 Jan 8; Accepted 2021 May 3
Copyright: ©2021 Yang et al.
Copyright year: 2021
Copyright holder: Yang et al.
License: This is an open access article distributed under the terms of the Creative Commons Attribution License, which permits unrestricted use, distribution, reproduction and adaptation in any medium and for any purpose provided that it is properly attributed. For attribution, the original author(s), title, publication source (PeerJ) and either DOI or URL of the article must be cited.
License URL: https://creativecommons.org/licenses/by/4.0/

Keywords: Sonneratia caseolaris, Chilling stress, RNA-Seq, Differentially expressed genes

Funding: Hainan Natural Science Foundation 319QN213 National Natural Science Foundation of China 31960281 32071503 This study was supported by grants from Hainan Natural Science Foundation (Grant No.319QN213) and the National Natural Science Foundation of China (Grant No. 31960281, 32071503). The funders had no role in study design, data collection and analysis, decision to publish, or preparation of the manuscript.

==============================
Sonneratia caseolaris is a native mangrove species found in China. It is fast growing and highly adaptable for mangrove afforestation, but suffered great damage by chilling event once introduced to high latitude area. To understand the response mechanisms under chilling stress, physiological and transcriptomic analyses were conducted. The relative electrolyte conductivity, malondialdehyde (MDA) content, soluble sugar content and soluble protein content increased significantly under chilling stress. This indicated that S. caseolaris suffered great damage and increased the levels of osmoprotectants in response to the chilling stress. Gene expression comparison analysis of S. caseolaris leaves after 6 h of chilling stress was performed at the transcriptional scale using RNA-Seq. A total of 168,473 unigenes and 3,706 differentially expressed genes (DEGs) were identified. GO and KEGG enrichment analyses showed that the DEGs were mainly involved in carbohydrate metabolism, antioxidant enzyme, plant hormone signal transduction, and transcription factors (TFs). Sixteen genes associated with carbohydrate metabolism, antioxidant enzyme, phytohormones and TFs were selected for qRT-PCR verification, and they indicated that the transcriptome data were reliable. Our work provided a comprehensive review of the chilling response of S. caseolaris at both physiological and transcriptomic levels, which will prove useful for further studies on stress-responses in mangrove plants.

Introduction

The growth, development, and geographical distribution of plants are limited by temperatures in their habitat (Jin & Kim, 2013). Low temperature stress can be divided into chilling stress (<10 °C) and freezing stress (<0 °C) (Zhou et al., 2011). Plants have evolved a series of response mechanisms to adapt to low-temperature environments. These include regulation of gene expression and physiological and biochemical process changes that enhance cold resistance (Thomashow, 1999; Sung et al., 2003), These changes have been studied in model plants like Arabidopsis thaliana and some cereal crops (Buti et al., 2018; Zhang et al., 2004; Román et al., 2012). Response mechanisms enhancing cold resistance mainly include signaling cascade, modification of the membrane lipid composition, accumulation of osmoprotectants, and increased activity of antioxidant enzymes to scavenge reactive oxygen species (ROS) (Theocharis, Clément & Barka, 2012; Nievola et al., 2017; Nakashima, Ito & Yamaguchi-Shinozaki, 2009). In addition, phytohormones such as auxin, abscisic acid (ABA), ethylene (ET), jasmonic acid (JA), salicylic acid(SA), gibberellin(GA), cytokinin(CK), and brassinosteroid (BR) and their signal transduction related genes also involved in regulating the gene expression in response to low-temperature stress (Peleg & Blumwald, 2011). Exogenous ABA improved the cold tolerance of Brassica napus; some members of WRKYs and MYBs transcription factors (TFs) regulated gene expression to participate in plant cold resistance (An et al., 2020; Marè et al., 2004).

Mangroves are halophytic temperature sensitive shrubs or trees that grow in the intertidal areas of tropical and subtropical coasts. They include about 70 species from 20 plant families (Duke, 1992). Mangroves are important in carbon balancing, protecting coastlines and beaches from floods and storms, reducing soil erosion, and providing habitats and shelters for animals. They also provide food, wood, and medical compounds for humans (Duke, 1992; Tomlinson, 1999; Schmitt & Duke, 2015). Large areas of mangroves, however, have been converted to agriculture, aquaculture, and industrial use (Duke et al., 2007; Robertsen & Alongl, 1992; Wang & Wang, 2007). There are approximately 22,684 ha of mangroves in China, which is 44% of the area existing in the 1950s (survey by State Forestry Administration in 2001). Since 2001, the government of China has focused on mangrove protection and restoration using conservation measures and large-scale afforestation in southeast China. The mangrove area in 2019 had increased to 29,000 ha. Sonneratia caseolaris is a mangrove species that is naturally distributed in the Hainan islands. Because of its rapid growth and adaptability, it was introduced to more northern and colder areas in China, such as Shenzhen and Leizhou in Guangdong Province, Qinzhou and Fangchenggang in Guangxin Province in 1994, where it grows well (Wang et al., 2002; Zan et al., 2003). S. caseolaris suffered substantial chilling injury in 2008, and there was heavy defoliation in all growing areas. All of the trees in Guangxi Province died. There was also significant defoliation of trees in native locations of Wenchang. Young seedlings were the most susceptible growth stage and suffered the great injury (Chen et al., 2017). Mangrove species with greater tolerance to low temperature are needed for mangrove restoration. Understanding the molecular mechanism of S. caseolaris tolerance to chilling stress will be of great value in restoration efforts.

Next-generation sequencing (NGS) is an efficient and inexpensive method for collecting transcriptome data. It can be used for species that have not been sequenced. Based on RNA-Seq platforms, genome-scale transcriptome analyses have been used to identify the mechanism of stress response without whole-genome sequence information in different species (He et al., 2015; Kumar et al., 2015). Transcriptome analysis has been used to study adaptive mechanism of mangroves, such as Acanthus ilicifolius, Avicennia offcinalis, Bruguiera gymnorhiza, Aegiceras corniculatum and Kandelia obovata (Fang et al., 2016; Krishnamurthy et al., 2017; Su et al., 2019; Yamanaka, Miyama & Tada, 2009; Yang et al., 2015a; Yang et al., 2015b). The chilling stress mechanisms of S. caseolaris, however, remains unknown.

To understand the mechanisms of S. caseolaris under chilling stress, we first measured the physiological changes in S. caseolaris under chilling stress, Then, we used RNA-seq for comparative analyses of S. caseolaris leaves between a control group (0 h, CK) and a chilling treated group (4 °C 6 h, CT). The reliability of the transcriptome results was verified by real-time quantitative polymerase chain reaction (qRT-PCR). The data improved our understanding of the chilling response mechanism in S. caseolaris and identified potential cold tolerance gene targets in this, and other mangrove species.

Materials and Methods

Plant materials and treatments

We collected ripe fruits of S. caseolaris in Hainan Dongzhaigang National Nature Reserve (19°57′23′′ N, 110°30′10′′ E) in China. The fruit was soaked in distilled water for 7 d and then individually planted in pots with a mud and sand (1:1) mixture as support. Growth chamber conditions were 30/28 °C (day/night) and a 14:10 h (L: D) photoperiod, 400 μ mol m−2 s −1 photon flux density, and 90% constant humidity. The germinated seeds were cultured in a growth chamber for 5 weeks, at which time the seedlings were about 12 cm high with 12 leaves. We performed the chilling stress treatment on these seedlings in a growth chamber, by setting the temperature to 4 °C and leaving the other conditions unchanged. The second leaf from the top of the seedlings were collected as samples from 0 h (CK) 3 h, 6 h, 12 h to 24 h separately, and then frozen in liquid nitrogen and stored at −80 °C. Each treatment had three biological replicates.

Physiological response assays of chilling-treated seedlings

We measured relative electrolyte conductivity following Wang et al. (2017). The malondialdehyde (MDA) content was measured by the method of Yao et al. (2011). The soluble sugar content was measured by the method of Wang et al. (2013). The soluble protein content was measured by using the Bradford method (Bradford, 1976). Three biological replicates were measured at each time point.

RNA extraction

Based on the physiological results, we used the seedlings without chilling as the control group (CK_1, CK_2, CK_3) and seedlings under chilling stress for 6 h were set as the treatment group (CT_1, CT_2, CT_3) for RNA-seq. Total RNA of leaf samples was extracted with TRIzol reagent (Invitrogen, Carlsbad, CA, USA) according to manufacturer instructions. RNA purity was checked by T6 Spectrophotometer (Puxi, Beijing, China). RNA integrity and concentration were measured by the RNA Nano 6000 Assay Kit of the Agilent Bioanalyzer 2100 system (Agilent Technologies, Santa Clara, CA, USA).

cDNA library construction and transcriptome sequencing

A 3 μg amount of total RNA of each sample was enriched using magnetic beads with Oligo (dT) and then fragmented into short fragments ∼200 bp by an interrupting reagent. The first strand cDNA was synthesized by a random hexamer primer, and the second strand of cDNAs was synthesized using the buffer, dNTPs, DNA polymerase I, and RNase H. After the double-stranded cDNA was generated and processed with end-repair, A-tailed, and adapters ligation, we constructed a cDNA library by PCR amplification. The cDNA library was then sequenced using the BGISEQ-500 platform with paired-end reads of 100 bp, by the Beijing Genomic Institution (www.genomics.org.cn, BGI, Shenzhen, China). All raw reads data from this study were submitted to the National Center for Biotechnology Information (NCBI) Sequence Read Archive database under the accession number PRJNA668053.

Transcripts assembly and functional gene annotation

The raw data were filtered with SOAPnuke (v1.5.2) (Li et al., 2008). We discarded the low quality reads such as those with adapter sequences, ambiguous bases ‘N’ is more than 5%, with more than 20% Q <10 bases. The remaining clean data were used for de novo assembly with the Trinity program (v2.0.6) (Grabherr et al., 2011) and then the unigene sequences were aligned with the following databases: NR, NT, SwissProt, KEGG, KOG, Pfam and GO by BLAST software (version 2.2.26).

Differential expression analyses

The expression level of each transcript was measured by RSEM (v1.2.12), and the expression level of a single gene was expressed as the value of FPKM (fragments per kilobase of transcript per million mapped reads) method (Li & Dewey, 2011). For the sake of determining the differentially expressed genes (DEGs), we performed differential expression analysis by the DESeq2 (Anders & Huber, 2010). A P-value ≤ 0.05 and an expression level —log2(ratio)—>2 were regarded as significant differential expression. Gene ontology (GO) enrichment analysis of DEGs was conducted using the GOseq R package, GO terms with p  <  0.05 were regarded as significantly enriched in DEGs (Young et al., 2010). We analyzed the KEGG pathway functional enrichment of DEGs using the R package. Significant enrichment was defined as when the FDR terms were < 0.001.

Quantitative real-time PCR analysis

We selected 16 DEGs to verify the reliability of transcriptome sequencing through qRT-PCR. The specific primers of qRT-PCR were designed by Primer3 software (http://primer3.ut.ee/) and are listed in Table S1. The qRT-PCR analyses were conducted using a LightCycler 480 RT-PCR system (Roche Applied Science, Mannheim, Germany) under the following conditions: 95 °C for 3 min, 40 cycles of 95 °C for 15 s, 56 °C for 30 s, and 72 °C for 20 s. Three replicates of each cDNA sample were performed for qRT-PCR analysis. We calculated the relative expression of each gene by the 2−DDCt method (Livak & Schmittgen, 2001) and used the actins gene of S. caseolaris as a reference.

Results

Physiological responses of S. apetala seedlings under chilling stress

After 3 h exposure to 4 °C, the leaves at the top of S. caseolaris seedlings began to show a slight curling, and after 6 h, the leaves in the top showed evident curling. After 12 h of chilling stress treatment, part of leaves had softened and exhibited loss of strength. After 24 h of 4 °C chilling stress, most of leaves had withered (Fig. 1). Electrolyte leakage, MDA content, soluble sugar content and soluble protein content increased significantly (Fig. 2). The electrolyte leakage increased from 6.54 ± 0.46%(0 h) to 28.56 ± 5.32% (24 h); the MDA content increased from 2.15 ± 0.79 µg/g mol/g (0 h) to 14.01 ± 1.58 µg/g mol/g (24 h); the soluble sugar contents increased from 2.79 ± 0.76 µg/g mol/g (0 h) to 7.87 ± 1.67 µg/g mol/g(24 h); and the soluble protein content increased from 6.00 ± 0.84 µg/g mol/g (0 h) to 33.83 ± 2.27 µg/g mol/g (24 h). All of these parameters showed a linear increase. When exposed to chilling stress for 6 h, these parameters increased 118.68%, 299.98%, 160.67% and 56.15% compared with the control group. On the basis of these results, we selected the leaf samples exposed to 0 h and 6 h of chilling stress for RNA-seq to investigate the changes in genome-wide gene expression mechanism under chilling stress. The 0 h time the control (CK) and 6 h was regarded as the treatment group (CT), with three biological replicates.

Figure 1 Phenotypic changes of S. caseolaris under chilling stress (4 °C).

Figure 2 Physiological change of S. caseolaris in chilling stress.

Error bar indicates ± SD of three replicates.

Sequencing and de novo assembly of transcriptome

In total, we obtained 39.33 Gb clean bases from 6 cDNA libraries, and we obtained a total of 262.21 M clean reads from 280.46 M raw reads (Table S2). The clean reads were de novo assembled by Trinity software and they yielded 168,473 unigenes. All unigenes were longer than 200 bp and ranged from 297 to 16,287 bp. The 200-300 bp unigenes accounted for the largest proportion (13,159, 12.26%) (Fig. S1A). The mean unigene length was 1,762 bp. The N50 and N90 lengths were 2,849 bp and 937 bp respectively, and the GC content was 43.94%.

Gene annotation and functional classification

To predict the putative function of the assembled unigenes, we used BLAST to compare seven databases including NR, NT, SwissProt, KEGG, KOG, Pfam and GO. The E-value cut-off was 10−5. A total of 125,360 (74.41%) unigenes were predicted to be homologous with known genes from the databases listed earlier. Among them, 116,619 (69.22%), 85,761 (50.90%), 90,660 (53.81%), 94,737 (56.23%), 100,382 (59.58%), 96,869 (57.50%) and 92,876 (55.13%) unigenes were found in the seven databases. A total of 48,858 (29.00%) were found in all seven databases (Table S3). The NR database had the highest proportion of annotations, with 69.22% unigenes being annotated. In the NR database, the unigene sequences exhibited the most similar BLASTx matches to gene sequences from Punica granatum (84,763, 72.69%), followed by Eucalyptus grandis (4,627, 3.97%) amd Quercus suber (1125, 0.96%) (Fig. S1B).

We classified the functions of the S. caseolaris unigenes by GO analysis and categorized 92,876 of the unigenes into 44 functional groups. These groups were classified into the following three major GO categories: 18 functional groups in “biological processes”, 12 functional groups in “cell component”, and 14 functional groups in “molecular function”. The dominant subcategories of the classified genes included “metabolic process (9,916)”, “cellular process (22,187)”, and “biological regulation (10,062)” in the “biological processes” category; “membrane part (29,068)”, “cell (23,014)”, and “organelle part (9,821) ” in the “cell component” category; “binding”(46,922), “catalytic activity (44,729)” and “transporter activity (5,902)” in the “molecular function” category (Fig. S1C). To analyze the intracellular metabolic pathways and biological functions in S. caseolaris, we conducted a KEGG analysis using the KEGG pathway tool. Genes were annotated to five categories: One functional group in “Cellular Processes”, two functional groups in “Environmental Information Processing”, three functional groups in “Genetic Information Processing”, 11 functional groups in “Metabolism”, and one functional group in “Organismal Systems” (Fig. S1D).

GO term enrichment analysis of DEGs

We identified a total of 3,706 DEGs, including 2,501 that were up-regulated and 1,205 that were down-regulated under the chilling stress (Fig. S2). The biological function information for the DEGs was analyzed by GO enrichment. A total of 2,902 DEGs were annotated in 1,411 terms and a total of 125 GO terms were identified. The most enriched GO categories among these DEGs were “nucleus”, “DNA binding”, “metal ion binding”, “DNA-binding transcription factor activity”, and “regulation of transcription, DNA-templated”, with 366, 239, 204, 128, and 125 DEGs respectively (Fig. 3A).

Figure 3 The top 20 GO and KEGG enrichment scatter diagram depicting the differential expression of genes.

(A) The top 20 GO enrichment. (B) KEGG enrichment.

Pathway enrichment analysis of DEGs

To characterize the active biological pathways of the DEGs in S. caseolaris under chilling stress, we classified the DEGs in KEGG biological pathways and conducted enrichment analysis. There were 2,557 DEGs annotated in 125 pathways. Among them, a total of nine KEGG pathways were significantly enriched with a P-value ≤ 0.05. The three most highly represented pathways were “Plant hormone signal transduction (ko04075)”, “Plant-pathogen interaction (ko04626)”, and “Glycerophospholipid metabolism (ko00564)”, with 115, 88 and 41 DEGs respectively (Fig. 3B). The largest number of DEGs included “plant hormone signal transduction”, indicating that plant hormones play an important role in response to chilling stress in S. caseolaris. The second largest number of DEGs included “Plant-pathogen interaction”, which suggested that low temperature exposed plants are more vulnerable to be pathogen infection. The “Glycerophospholipid metabolism” category had the third largest number of DEGs, suggesting that fatty acid metabolism responded positively to low temperature. A considerable number of biological reactions were altered in S. caseolaris in response to chilling stress.

DEGs associated with chilling stress

Carbohydrate metabolism related DEGs

Carbohydrates played an important role in plant response to low temperatures. In GO enrichment analysis, 12 DEGs were enriched in “starch binding (GO:2001070)” and all were up-regulated. Many DEGs that were involved in starch, fructose, sucrose, glucose, and trehalose were up-regulated in chilling stress (Fig. 4A). Four genes (CL1250.Contig10_All, CL13708.Contig9_All, CL3523.Contig11_All, and CL5529.Contig9_All) were involved in trehalose biosynthetic process, two sucrose synthase genes (CL14150.Contig25_All and CL2626.Contig4_All), and one sucrose-phosphate synthase gene (CL2626.Contig4_All) were involved in the sucrose process. Many DEGs were involved in “Pentose phosphate pathway (ko00030)”, “Fructose and mannose metabolism (ko00051)”, “Galactose metabolism (ko00052)”, “Pentose and glucoronate interconversions (ko00040)”, and “Starch and sucrose metabolism (ko00500)” respectively, and most of these were up-regulated.

Figure 4 Heat map of relative expression levels of DEGs involved in main transcription factors, sugar metabolism and the antioxidant defense system.

(A) The main sugar metabolism-related genes. (B) The main antioxidant-related genes. (C) The main transcription factors.

Phytohormone related DEGs

KEGG analysis of DEGs in S. caseolaris showed that hormone signal transduction was significantly influenced by chilling stress (Fig. 3). There were 115 DEGs enriched in “plant hormone signal transduction” and most of them were up-regulated (Fig. 5). Especially in auxin signal transduction, ABA signal transduction, ET signal transduction and JA signal transduction.

In this GO analysis of DEGs, 41 DEGs enriched in the “auxin-activated signaling pathway (GO:0009734)” term. Among them, AUX1 (Unigene13181_All) is a key gene that encodes an auxin receptor, and down-regulated 2.50 fold. In the ABA signal pathway, PLY and PP2C are key enzymes controlled by the ABA signaling pathway. In this study, the genes encoding these two enzymes (CL1229.Contig3_All, Unigene14664_All, and CL5508.Contig1_All) were all up-regulated. Additionally, seven DEGs were enriched in the “abscisic acid-activated signaling pathway (GO:0009738)” term. In the ET signal pathway, we found that the one gene (CL2232.Contig4_All) encoding ETR down-regulated 6.81 fold. In the JA signal pathway, cetyl-CoA acyltransferase 1 is a key enzyme for synthesis of JA-CoA. In this study, the genes (CL10456.Contig6_All) encoding this enzyme were up-regulated by 3.93-fold.

Figure 5 Plant hormone signal transduction and the relative heat map.

(A) Plant hormone signal transduction. (B) Heat map of DEGs involved with auxin, ABA, ET and JA.

ROS related DEGs

We found DEGs related to catalase, glutathione peroxidase, peroxidase, and superoxide dismutase were involved in chilling stress (Fig. 4B). Two genes encode superoxide dismutase (CL10965.Contig3_All and CL355.Contig5_All). 3 genes encode catalase (Unigene30041_All, Unigene30063_All and Unigene30085_All), and 10 genes encode peroxidase (CL1068.Contig11_All, CL1068.Contig6_All, CL1068.Contig7_All, CL11425.Contig1_All, CL13638.Contig5_All, CL2936.Contig3_All, CL3639.Contig5_All, CL3639.Contig6_All, CL4125.Contig8_All, and Unigene3189_All). These genes were all up-regulated, indicating that they may have played a positive role in activating antioxidant enzymes in S. caseolaris exposed to low temperatures.

Figure 6 (A-Q) Expression of S. caseolaris genes in response to chilling stress for 0 h (CK) and 6 h (CT) as determined by qRT-PCR (blue) and RNA-seq (red).

Transcription factors

TFs play an important role in regulating gene expression that participates in plant stress regulatory processes, such as chilling stress-related processes. In this study, we classified a total of 232 TFs into 41 families and showed differential expression in chilling treated seedlings. A total of 161 (69.40%) were up-regulated. The five most abundant TF families were AP2-ERF, MYB, C3H, WRKY, and ARF with 35 (15.09%), 29 (12.50%) 21 (9.05%), 21 (9.05%), and 15 (6.47%) respectively (Fig. 4C).

Validation of transcriptome data by RT-qPCR analyses

To validate the RNA-Seq results, we selected a total of 16 candidate genes for qRT-PCR analyses (Fig. 6). Based on enrichment analyses, all of these genes were involved in the chilling response. They represented different functional categories or pathways, such as sugar metabolism, antioxidant enzyme, phytohormones and TFs. The results indicated that the transcriptome results were reliable.

Discussion

With its rapid growth adaptability, S. caseolaris is a good candidate species for mangrove afforestation in China. Low temperature, however, have limited its distribution in high latitude areas. Understanding the physiological and molecular mechanisms of chilling resistance can aid in breeding cold tolerant varieties of S. caseolaris. We performed physiological and transcriptomic analyses of leaves under 4 °C chilling stress, and then analyzed the DEGs involved in the antioxidant defense system, carbohydrate metabolism, plant hormone signals, and TFs.

Membrane injury of S. caseolaris seedlings caused by chilling stress

Membrane lipids are the first place that plants perceive and respond to abiotic stress signals such as low temperature. Lipids are involved in biological process, such as signal transduction, energy conversion, and metabolic regulation (Xie et al., 2018). Damaged membrane lipids increase electrolyte leakage, and a high electrolyte leakage rate reflects serious damage caused by chilling stress. Membrane permeability is an important physiological index of plant resistance to stress. MDA is the final degradation product of lipid peroxidation, and the MDA level reflects the damage of membrane lipid peroxidation and the stress tolerance of plant cells (Mutlu, Karadaolu & Nalbantolu, 2013; Wise & Naylor, 1987). Physiological analysis showed that the relative electrolyte conductivity and the MDA level both increased significantly after chilling stress. Membrane lipid peroxidation likely occurred and membrane lipids were damaged. Similar results were found in Magnolia wufengensis (Deng et al., 2019). Serious member lipid injury can result in seedling death. This might explain why most seedlings died when exposed to extreme low temperatures (Chen et al., 2017). We found that DEGs were enriched in “Glycerophospholipid metabolism (ko00062)” and “Fatty acid elongation (ko00564)”. The genes responded to the increased level of unsaturated fatty acids and membrane fluidity in the cells of S. caseolaris seedlings under chilling stress, which may be a mechanism of cellular self-repair.

Carbohydrate metabolism-related DEGs

When exposed to low temperatures, plants usually break down starch and increase the content of carbohydrates to maintain stability of cell membrane structure, this could increase the low temperature tolerance of plant cells (Guy et al., 2008). In GO enrichment analysis, 12 DEGs were enriched in “starch binding (GO:2001070)” (Fig. 3B). This could result in the conversion of starch into soluble sugar under chilling stress. Physiological experiments confirmed that the soluble sugars content of S. caseolaris increased after chilling stress. Increased levels of sucrose, raffinose, glucose, and fructose are involved in the low-temperature resistance of other plants (Anchordoguy et al., 1987; Ma et al., 2009). The RNA-seq data showed that many DEGs that are involved in starch, fructose, sucrose, glucose, and trehalose were up-regulated in chilling stress treatment. These data were consistent with previous studies (Wang et al., 2018; Deng et al., 2019; Yang et al., 2019). Plants can accumulate trehalose for oxidative detoxification to counteract abiotic stresses (Lai et al., 2017). In the DEGs, four genes were involved in the trehalose biosynthetic process and all of them were up-regulated, which is similar to the report of Deng et al. (2019). Sucrose synthase and sucrose phosphate synthase play important roles in sucrose metabolism and biosynthesis respectively (Winter & Huber, 2000). We found that two sucrose synthase genes and one sucrose-phosphate synthase gene were up-regulated. These genes may have been involved in the response to chilling stress. A similar result was found by Yang et al. (2019). Carbohydrate also regulate expression of jasmonate, ABA and COR genes (Rutten & Santarius, 1992; Welling & Palva, 2006). We found that many DEGs were involved in carbohydrate metabolism, which indicated that carbohydrates play an important role in S. caseolaris response to chilling stress.

Antioxidant defense system

Under environment stress, ROS such as superoxide, hydroxyl radicals and hydrogen peroxide can accumulate in plant cells and cause oxidative damage (Elsheery & Cao, 2008; Suzuki & Mittler, 2006). To scavenge ROS, antioxidant systems such as superoxide dismutase (SOD), catalase (CAT), peroxidase (POD), and ascorbate peroxidase (APX) are activated in response to plant stress (Mittler, 2002). SOD is the first antioxidant enzyme that could convert O2− to hydrogen peroxide (H2O2), and then POD and CAT could catalyze H2O2 and convert it to H2O. We found DEGs related to CAT, glutathione peroxidase, POD, and SOD were up-regulated, indicating that they may have played a key role in activating antioxidant enzymes in S. caseolaris exposed to low temperatures. ROS are also signal molecules that can induce gene expression and protein synthesis to increase stress tolerance in plants (Jaspers & Kangasjärvi, 2010). ROS also is active the mitogen-activated protein kinase (MAPK) cascade pathway, which regulates low-temperature response genes involved in TFs and some hormones (Pitzschke, Schikora & Hirt, 2009; Xie et al., 2009). We found 15 DEGs enriched in “MAP kinase activity” and most of them were un-regulated, which indicated that MAPK signal transduction pathway also played an important role in chilling stress responses.

These results indicated that many antioxidant genes are activated, and these would not only increase the activity of antioxidant enzymes to scavenge with ROS, but also active a sequence of signal to response chilling stress.

Phytohormone signal network responses to chilling stress

Phytohormones are the secondary signals in plant cells that play a significant role in adaptation to environmental stress (Peleg & Blumwald, 2011). They can activate a sequence of signal events and eventually induce the genes that directly respond to stress (Bari & Jones, 2009; Hakeem, Rehman & Tahir, 2014). We found four key response hormone signal transduction pathways (auxin, ABA, ET, and JA) that were significantly influenced by chilling stress. Auxin is a plant growth-promoting hormone (Nemhauser, Hong & Chory, 2006). In this study, 41 DEGs were enriched in the “auxin-activated signaling pathway (GO:0009734)” term. Down-regulation of the AUX1 gene may reduce plant growth in response to chilling stress. The ABA signaling pathway is important in regulating abiotic stress responses in plants (Danquah et al., 2014). It can regulate stomatal closure, reduce water loss and maintain cell growth (Peleg & Blumwald, 2011). PLY and PP2C are key genes controlled by the ABA signaling pathway. In present study, the genes were up-regulated in response to chilling stress. This could activate high expression of PP2C and promote stomatal closure and decrease transpiration to reduce water loss. Similar results have been found on Arabidopsis thaliana (Tähtiharju & Palva, 2001) and Oryza sativa L (Guan et al., 2019). Additionally, the DEGs were also enriched in the “abscisic acid-activated signaling pathway (GO:0009738)” term, which further confirmed that ABA-dependent signaling is involved in the low temperature response of S. caseolaris. Genes related to ET biosynthesis often down-regulated expressed under low temperature treatment (Chu & Lee, 1989). In this study, the key gene encoding ETR was down-regulated, which was consistent with previous reports (Yang & Huang, 2018). JA plays a significant role in plants responding to abiotic and biotic stress (Wasternack & Song, 2017; Wasternack & Strnad, 2018). The up-regulated key genes may promote the synthesis of JA-CoA and promote the synthesis of JA. Exogenous methyl salicylic acid (MeSA) or methyl jasmonate (MeJA) could induce the transcription of pathogenesis-related genes, and increase chilling tolerance and pathogen resistance (Ding et al., 2002). This may explain why many DEGs were enriched in “Plant-pathogen interaction”. Similar results have been found in K. obovata, M. wufengensis, and Camellia sinensis under chilling stress (Deng et al., 2019; Li et al., 2019; Su et al., 2019). However, how those plant hormones mediated signaling participates in chilling stress responses in S. caseolaris still need to be further study.

TFs responding to chilling stress

TFs play important roles in plant growth and stress tolerance (Lee, Henderson & Zhu, 2005). When a plant experiences the chilling stress, a series of signal transduction pathways could activate TFs. The activated TFs will activate the expressions of a cascade of downstream resistance-related genes to enhance the cold tolerance by special binding (Kasuga et al., 1999). Many TF families such as DREB (Ahmed et al., 2017; Chen et al., 2012), MYB (Zhao et al., 2012; Agarwal et al., 2006; Chen et al., 2006), and WRKY (Zou, Jiang & Yu, 2010) are related to plants resistance to low temperature stress. We identified 232 TFs that were classified into 41 families. AP2-ERF, MYB, C3H, WRKY, and ARF were five major TF families identified as DEGs, suggest that they played a crucial role in response to chilling stress. AP2-ERF genes belong to a plant-specific TF family involved in regulating stress responses (Mizoi, Shinozaki & Yamaguchi-Shinozaki, 2012). The AP2-ERF superfamily consists of the ERF, RAV, AP2, and DREB families. Overexpression of AP2 TF, JcDREB, and BpDREB2 increased cold tolerance of Arabidopsis (Tang et al., 2011; Sun et al., 2014). Previous studies verified that AP2-ERFs involved in a lot of plant hormone mediated abiotic stress responses, such as ABA, ET, GA, CTK, and BR (Colebrook et al., 2014; Kazan, 2015; Tao et al., 2015; Nolan et al., 2017; Sah, Reddy & Li, 2016). MYB are involved in chilling stress and regulation of ABA-responsive genes, and constitute with ABA-dependent, induced and mediated types. Which plays an important role in the upstream stage of low temperature transduction (Gyoungju et al., 2016). AtMYB52, AtMYB70, and AtMYB82 were ABA-dependent genes in responding abiotic stresses of Arabidopsis (Park, Kang & Kim, 2011); TaMYB1, TaMYB2, TaMYB3R1 were ABA-induced genes in responding abiotic stresses of wheat (Cai et al., 2011; Tong et al., 2006); AtMYB2, AtMYB41 were ABA-mediated types that promote the accumulation of ABA in responding abiotic stresses (Abe et al., 2003; Lippold et al., 2009). The WRKY genes can respond strongly and rapidly to abiotic stresses (Chen et al., 2012). The WRKY46 gene enhanced the low temperature tolerance of transgenic cucumber by activating a series of cold-stress responsive genes in an ABA-dependent manner (Zhang et al., 2016). MaWRKY25 can be induced by low temperature or MeJA, which could increase the low temperature tolerance of banana. It could also active the MaLOX2, MaAOS3, MaOPR3 genes to promote the synthesis of JA by special binding. Which could reduce the damage of banana fruit by low temperature (Ye et al., 2016). Other TFs, such as the HSF, bZIP, and GRAS families, are induced by chilling stress, and members of these families can affect the cold resistance of plants (Chinnusamy, Zhu & Sunkar, 2010). These results indicated that many TFs were involved in the seedlings of S. caseolaris in response to chilling stress through different pathways. Besides, the both un-regulated TFs and phytohormones maybe have some connection, and the regulatory network need further study.

Figure 7 Hypothetical model of the events occurring in the S. caseolaris leaves under chilling stress.

Conclusions

We analyzed at the morphological, physiological and transcriptome level of S. caseolaris response to chilling stress. Chilling stress of S. caseolaris likely involves damage to the cell membranes. This was supported by a significant increase in the relative electrolyte conductivity and MDA contents. Up regulating genes in chilling stressed S. caseolaris were involved in plant hormone signal transduction, TFs, carbohydrate metabolism, and antioxidant enzymes. These genes might be involved in the significant increase of soluble protein content and soluble sugar content in response to the chilling stress (Fig. 7). These data revealed the manner in which S. caseolaris responds to chilling stress, and provided insight for further research on the environmental adaptations of mangrove species.

Supplemental Information

Supplemental Information 1 Characteristics of unigenes

(A) Distribution of unigene lengths in Sonneratia caseolaris. (B) Main species distribution of S. caseolaris unigenes. (C) Pathway annotation of unigenes based on KEGG categorization (D) Functional annotation

Click here for additional data file.

Supplemental Information 2 Volcano diagram showing the differential expression of genes. Blue represents down-regulated, red indicates up-regulated

Click here for additional data file.

Supplemental Information 3 List of primers used in this study

Click here for additional data file.

Supplemental Information 4 Summary of sequence analysis

Click here for additional data file.

Supplemental Information 5 Number of annotated unigenes

Click here for additional data file.

Supplemental Information 6 Raw data of Fig. 2

Click here for additional data file.

Supplemental Information 7 Raw data of Fig. 5

Click here for additional data file.

Additional Information and Declarations

Competing Interests

Author Contributions

Data Availability

The authors declare there are no competing interests.

Yong Yang conceived and designed the experiments, performed the experiments, authored or reviewed drafts of the paper, and approved the final draft.

Chunfang Zheng, Ying Zhang and Qiang Liu conceived and designed the experiments, authored or reviewed drafts of the paper, and approved the final draft.

Cairong Zhong performed the experiments, prepared figures and/or tables, collected the samples, and approved the final draft.

Tianxi Lu and Xiang Jin analyzed the data, prepared figures and/or tables, and approved the final draft.

Juma Gul performed the experiments, prepared figures and/or tables, cultivated seedlings, and approved the final draft.

The following information was supplied regarding data availability:

The data is available at NCBI: PRJNA668053.

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
