# Peer review of "Transcriptome analysis of Sonneratia caseolaris seedlings under chilling stress"

_PeerJ, doi:10.7717/peerj.11506_

## Round 0.1 · original submission · Major Revisions

In view of the criticisms of the Reviewers found at the bottom of this letter and my own assessment, a revision of your manuscript is required for publication in PeerJ.

All reviewers agreed with what this work contains interesting findings, which are potentially satisfying the level for publication. Hence, I expect the authors to read the Reviewers' comments thoroughly and revise the manuscript to address their concerns.

In addition, the authors should deposit at least raw read data but also unigenes' sequences to a public database (to get the accession number (e.g. DRAXXXXX))  or website (alternatively make supplemental figs for the sequences of unigenes used in this paper) before the next submission to allow the readers to access the NGS data shown in this manuscript.  I think what the authors did not show such information is most serious flaw in this paper. At least the authors must show the accession number for raw read data in the manuscript. This is the rule for publication using NGS data.  Without such information, nobody can confirm your work.

Reviewer 1 ·

Basic reporting

The manuscript titled “Transcriptome Analysis of Sonneratia caseolaris Seedlings Under Chilling Stress” mainly focused on the response to chilling stress of Sonneratia caseolaris. Besides, the transcriptome data of the leaves under chilling stress was also studied. After treated by low temperature for 6h, there were 168473 unigenes and 3706 DEGs identified in leaves. And authors analyzed the relative pathways and their relationships on chilling response reactions. However, there are some issues should be improved in this manuscript.

Experimental design

1. During the chilling treatment, authors should add the thermoperiod, photoperiod and illumination intensity.
2. For the experimental design, how many seedlings were used in the treatment? And how many repetitions were used in the treatment?
3. In the section of Plant materials and treatments, why authors chose 4℃ as low temperature treatment? Please explain.
4. In the section of Plant materials and treatments, the second leaf of seedling was taken as samples. Please describe more detailed with the position of the sample leaf. And explain why choose this leaf as the sample.
5. In the test of qPCR, the author mentioned that “Three replicates of each cDNA sample were performed for qRT-PCR analysis”, how many technical duplication were used in the fluorescence quantitative step? In the initial data, I only saw 3 replicates data.

Validity of the findings

6. In the Results section, there is only one sentence to Figure 1. Please give a more detailed description on the changes during chilling stress.
7. In the Results section, the author didn’t describe Figure 4 B and C.
8. The author didn’t describe Figure 6 in the Results section.
9. For the pathway enrichment analysis, the Plant-pathogen interaction was explained as “low temperature exposed plants are more vulnerable to be infection by pathogens” by authors. The author should consult a large number of similar references and speculate the possible reason of this phenomenon in the Discussion section.

Additional comments

The manuscript was well performed, but there are some issues should be improved before published. So I recommend major revision.

·

Basic reporting

The RNA-seq data has interesting results. However, the authors did not critically analyse the data to hypothesize the mechanism of chilling stress in S. caseolaris.Hence, the authors need to improve the analysis part in more detail.

Experimental design

Research findings are not rigorous, rather very general. Since they have quite interesting results, they should analyse the mechanism of chilling stress in more detail and critically. Experimental design is fine. However, the auhtors need to do more qRT-PCR analysis for more transcription factors and few more for each transcription factor.

Validity of the findings

The results are interesting and has importance in the field as chilling stress mechanism in mangrove is not explored yet. However, they need to work on the data critically to come up with a hypothesis. The physiological part is fine, but the molecular mechanism part is not analysed in detail. Hence, the conclusion is very general.

Additional comments

The manuscript #56218 entitled "Transcriptome analysis of Sonneratia caseolaris seedlings under chilling stress” by Yang et al. aimed to investigate physiological and transcriptomic analyses to understand the mechanisms of chilling stress of Sonneratia caseolaris seedlings. The physiological analyses was performed on relative electrolyte conductivity, malondialdehyde (MDA) content, soluble sugar content and soluble protein content. They observed that the content of these compounds increased significantly under chilling stress. It suggests that the levels of osmoprotectants increased in S. caseolaris seedlings in response to the chilling stress. To better understand the mechanism, they performed RNA-Seq of S. caseolaris leaves after 6 h of chilling stress. The pathway analysis of DEGs showed that the highly enriched categories were related to carbohydrate metabolism, antioxidant enzyme, plant hormone signal transduction, and transcription factors (TFs). They have quite good and interesting data, but the analysis and presentation of the data needs major improvement.

I have few critical comments for the authors:

1. Introduction needs major revision. They need to add information on the physiological and molecular mechanism in response to chilling stress in other plants. I agree that it is not known in mangrove, but in general what happens to plants under chilling stress and what are the molecular mechanisms already established (including transcription factors and different phytohormones) that plants adapt to overcome the chilling stress need to be explained in the introduction to have a general overview.
2. The information on chilling stress in the first paragraph of discussion part should be in the introduction.
3. Discussion part needs critical revision.
4. Results are included in the discussion part. Hence, the authors should remove those parts and write that information in the results section so that discussion part will be shorter.
5. Phytohormones play a major role during abiotic stresses. The gene expression data on the phytohoromes seems quite interesting. I don’t see much analysis and explanation on hormonal regulation in the manuscript except the information on the up and down regulation of genes associated with hormones. It is very important to analyse and discuss on hormonal regulation by linking the hormone responsive TFs and that would help the authors to come up with some hypothesis regarding the molecular mechanism of chilling stress in S. caseolaris.
6. The authors have very interesting results on the gene expression data of different TFs. Many of them are strongly upregulated. However, I do not see any critical information on the role of transcription factors on chilling stress mechanism. The authors have just mentioned that they are up/down regulated and hence they are important. However, they need to explain the role of each TF in more detail with the possible regulatory mechanism involved. They need to also pay attention on any transcriptional changes happening during chilling stress. Please add a section on different TFs and their possible role in chilling stress and among them which of the potential TFs could be playing a key role in the chilling stress in S. caseolaris. To support their findings and hypothesis the authors need to do more qRT-PCR analysis for more number of TFs such as AP2/ERF, ARF, C2H2 ZnF, bHLH, MYB, WRKY etc. Instead of doing one transcript for each TF, please do it for more number of copies so that it will give more support to your findings. You can have a hypothetical model by linking the physiological and molecular mechanism. Overall, there is no critical analysis of the results.
7. The authors should have a summary figure which could explain their overall findings and hypothesis regarding the chilling stress mechanism.

---

## Round 0.2 · Minor Revisions

The authors seem to address all concerns raised by reviewers and me and revised the manuscript according to the comments from reviewers. However, one reviewer still has comments on the revised manuscript. Hence, I would like to wait for a proper revision before acceptance for publication.

·

Basic reporting

The authors have revised the manuscript. I do not have any comments as they have addressed all my comments.

Experimental design

The authors have done the suggested experiment. Hence, I do not have any comments.

Validity of the findings

They have validated their findings.

Additional comments

The authors have addressed all my comments. However, I have a doubt in the revised manuscript. In the results section, under sub heading "Phytohormone related DEGs" line 257-258, they have mentioned that
"Especially in auxin signal transduction, ABA signal transduction, EN signal transduction and JA signal transduction." What is EN signal transduction? Is it ethylene (ET) signal transduction pathway? Please check it and edit it using the correct wording in the manuscript and also in the figures.

---

## Round 0.3 · Minor Revisions

Before I accept the manuscript, Jasmine Janes, the Section Editor, has commented and said:

"Just a minor revision needed - what are the error bars on Fig 2? Are they SD, SE, CI? In the text, are the values for Fig 2 the mean across the three replicates with SD values? Or something else?"

I think this is a serious concern, but I confirmed that there is no explanation about it in the figure caption. Hence, I would like to ask the authors to add the caption to explain this point prior to acceptance.

---

## Round 0.4 · accepted · Accept

I confirmed that the authors addressed the additional concerns raised by the section editor. Hence, I think that the current version of the manuscript deserves publication in this journal.